# FOSB–PCDHB13 Axis Disrupts the Microtubule Network in Non-Small Cell Lung Cancer

**DOI:** 10.3390/cancers11010107

**Published:** 2019-01-17

**Authors:** Chen-Hung Ting, Kang-Yun Lee, Sheng-Ming Wu, Po-Hao Feng, Yao-Fei Chan, Yi-Chun Chen, Jyh-Yih Chen

**Affiliations:** 1Marine Research Station, Institute of Cellular and Organismic Biology, Academia Sinica, Ilan 262, Taiwan; koichiting@gmail.com (C.-H.T.); yotaka1225@gmail.com (Y.-C.C.); 2Graduate Institute of Clinical Medicine, College of Medicine, Taipei Medical University, Taipei 110, Taiwan; kangyunlee68@gmail.com; 3Division of Pulmonary Medicine, Department of Internal Medicine, Shuang Ho Hospital, Taipei Medical University, New Taipei City 235, Taiwan; chitosan@tmu.edu.tw (S.-M.W.); pohao.feng@gmail.com (P.-H.F.); k2201025@ms19.hinet.net (Y.-F.C.); 4Division of Pulmonary Medicine, Department of Internal Medicine, School of Medicine, College of Medicine, Taipei Medical University, Taipei 110, Taiwan; 5The iEGG and Animal Biotechnology Center, National Chung Hsing University, Taichung 402, Taiwan; 6Center of Excellence for the Oceans, National Taiwan Ocean University, Keelung 202, Taiwan

**Keywords:** antimicrobial peptide, mitochondrial stress, FOSB, protocadherin, cytoskeleton, non-small cell lung cancer (NSCLC)

## Abstract

Non-small cell lung cancer (NSCLC) is among the leading causes of human mortality. One reason for high rates of NSCLC mortality is that drug resistance is a major problem for both conventional chemotherapies and less-toxic targeted therapies. Thus, novel mechanistic insights into disease pathogenesis may benefit the development of urgently needed therapies. Here we show that FBJ murine osteosarcoma viral oncogene homolog B (FOSB) was induced by an antimicrobial peptide, tilapia piscidin-4 (TP4), through the dysregulation of mitochondrial Ca^2+^ homeostasis in NSCLC cells. Transcriptomic, chromatin immunoprecipitation quantitative PCR, and immunocytochemical studies reveal that protocadherin-β13 (*PCDHB13*) as a target of FOSB that was functionally associated with microtubule. Overexpression of either PCDHB13 or FOSB attenuated NSCLC growth and survival in vitro and in vivo. Importantly, downregulation of both FOSB and PCDHB13 was observed in NSCLC patients and was negatively correlated with pathological grade. These findings introduce the FOSB–PCDHB13 axis as a novel tumor suppressive pathway in NSCLC.

## 1. Introduction

Lung cancer is the leading cause of cancer mortality, and most patients with this disease are diagnosed with non-small cell lung cancer (NSCLC) [1]. Systemic chemotherapy remains the first-line treatment for the patients with NSCLC. However, chemotherapies affect healthy cells and cause multidrug resistance [2]. Patients harboring tumors with epidermal growth factor receptor (EGFR) mutations can be treated by EGFR tyrosine kinase inhibitors (TKIs) [3,4,5,6,7,8,9]. However, TKI treatment may be compromised by novel mutations, which create drug resistance [10]. Thus, characterization of other mechanisms involved in the NSCLC pathogenesis will be crucial for the development of novel therapies.

Cationic antimicrobial peptides (AMPs) are evolutionarily conserved and involved in the innate immune response combatting pathogens [11]. AMPs have been reported to selectively target cancer cells with lower toxicity to normal cells and thus may serve as potent anticancer drugs [12,13,14,15]. The selective toxicity to cancer cells is probably due to the negative charge often found on the plasma membranes of cancer cells [12,14,16]. Strong cytotoxicity of TP4 was previously observed in NSCLC cells but not in normal cells [17]. Extensive glycosylation and low cholesterol levels are found on lung cancer cell membranes are not found in normal lung cells and may enhance the negative charge of the plasma membrane on the cancer cells [18,19]. According to previous studies, the mechanism of action underlying AMP-mediated cancer toxicity may involve the induction of activator protein-1 (AP-1) [13,14] and cytoskeleton disruption [15]. AP-1 members are possible oncogenic or tumor suppressive factors, depending on the cellular context and the genetic background of the tumor [20]. We have reported that the Nile tilapia (*Oreochromis niloticus*)-derived cationic AMP—Tilapia piscidin-4 (TP4)—is highly toxic to cancer cells [14,15,17]. TP4 induces the AP-1 subunit, FOSB, of which expression is negatively associated with cell differentiation status and pathological grade in breast cancer [14,21,22]. In addition, FOSB overexpression triggers cancer cell death [14], suggesting a tumor-suppressive role of the protein. It was shown previously that FOSB is significantly downregulated in NSCLC [23]; however, its role in disease progression remains to be characterized.

AP-1 is known to regulate cadherins, which function as tumor suppressors or proto-oncogenic proteins via their effects on cell polarity and cellular reprograming [24,25,26,27]. A related class of molecules, protocadherins (PCDHs), has also been implicated in tumorigenesis [28,29,30,31,32,33]. Previous studies showed that DNA methylation of the *PCDHB* cluster can serve as a marker of aggressiveness in malignancies [31,32]. Importantly, most genes in the *PCDHB* cluster exhibit reduced expression in NSCLC [34], but it remains unclear how PCDHB genes function during tumorigenesis, and whether their function requires AP-1.

Here we show that FOSB was induced by TP4 via mitochondrial damage-triggered Ca^2+^ dysregulation, leading to NSCLC cell death. Transcriptomic analysis revealed that FOSB activation disrupted cytoskeletal and membrane integrity in NSCLC cells. We also found that FOSB transcriptionally activates *PCDHB13*, which is functionally associated with microtubules. Importantly, elevated expression of FOSB and PCDHB13 diminished cell survival in vitro and in vivo. Furthermore, the expression levels of both FOSB and PCDHB13 were negatively correlated with the clinical status of NSCLC patients, suggesting that the FOSB–PCDHB13 axis may be a promising molecular target for the NSCLC.

## 2. Results

### 2.1. Mitochondrial Stress-Induced FOSB Expression Requires Ca^2+^ Signaling

We have previously shown that AMP induces stress-inducible AP-1 transcription factors and triggers cancer cell death [13,14]. Thus, we examined AP-1 levels in NSCLC cells upon TP4 treatment. TP4 was previously shown to cause broad cytotoxicity to NSCLC cells at doses over 3.35 μM [17]. FOSB protein level was increased after TP4 treatment—but not c-JUN, JUNB, JUND, cFOS (Figure 1A and Appendix A), FRA1, or FRA1-associated proteins (SNAIL and MMP2) (Appendix A). Induction of FOSB was of particular interest, due to its known tumor suppressive effects [14,21,22] and because the induction occurred in lung epithelial cancer cells but not in the control cell line, a normal human bronchial epithelial line, BEAS-2B (Figure 1A). We subsequently explored FOSB-mediated effects in NSCLC. The viability of NSCLC cells was determined after manipulating expression of FOSB and its alternatively spliced form, FOSΔB. Overexpression of either FOSB or FOSΔB could induce cell death (Figure 1B), while knockdown of FOSB/FOSΔB (Figure 1C,D) conferred resistance to TP4 (Figure 1E), indicating that FOSB is involved in TP4-induced death. To explore the mechanism of TP4-induced FOSB-mediated cytotoxicity in NSCLC, we first examined TP4 localization by immunocytochemistry (ICC). The result shows that TP4 was colocalized with Prohibitin, which is known to regulate cristae morphogenesis of mitochondria (Figure 1F, upper panel). TP4 was not colocalized with giantin, which forms the intercisternal cross-bridges of the Golgi, nor was it colocalized with calreticulin, which acts as a Ca^2+^-binding chaperone in the ER lumen (Figure 1F, middle and lower panels, respectively). The mitochondrial targeting of TP4 was confirmed by probing for TP4 in the mitochondrial fraction with Western blotting (Figure 1G). We then asked whether the mitochondrial targeting may affect mitochondrial polarization. The activity of mitochondria can be inferred by the membrane potential-dependent accumulation of MitoTracker. Organelle morphology was obviously altered, and the polarization was significantly affected by TP4 at toxic concentrations (5.03 and 6.71 μM) (Figure 1H). Since mitochondria regulate Ca^2+^ to influence cell metabolism and death, we examined Ca^2+^ homeostasis. Rhod-2 AM staining revealed depletion of mitochondrial Ca^2+^ in cells receiving high doses of TP4 (5.03–20.12 μM) but not in cells receiving mild to moderate doses (0.84–3.35 μM) (Figure 1I). Fluo-4 intensity further indicated that the cytoplasmic Ca^2+^ level was significantly increased after effective doses of TP4 treatment (≥3.35 μM, Figure 1J).

Maximal FOSB induction was observed at 3 h post-TP4 treatment, after the increase in cellular Ca^2+^ concentration and in concert with increased ERK phosphorylation (Figure 1K; Appendix A). This timing led us to investigate whether FOSB induction requires Ca^2+^. Pretreatment of cells with the Ca^2+^ chelator, BAPTA/AM, prevented FOSB induction (Figure 1L) and cell death (Figure 1M). In addition to Ca^2+^, the activity of AP-1 often requires ERK/JNK signaling [35]. We therefore blocked the ERK/JNK pathway and tested TP4-induced effects. However, ERK/JNK blockade with PD98059 or JNK inhibitor VIII in A549 cells efficiently induced cell death on its own (Appendix A), indicating that ERK/JNK signaling is essential for NSCLC cell survival. Together, these findings show that mitochondrial stress induces TP4-triggered FOSB expression in a Ca^2+^ dependent manner.

### 2.2. FOSB Regulates Cellular Integrity in NSCLC

To examine how FOSB induction causes cytotoxicity, we conducted transcriptome analysis of FOSB-overexpressing cells. The results of a gene ontology (GO) analysis comparing FOSB- and EGFP-overexpressing cells showed that 54% of differentially expressed genes were associated with the membrane and cytoskeleton (Figure 2A,B and Appendix A). Thus, we hypothesized that FOSB upregulation may induce morphological and cytoskeletal changes. Indeed, the microtubule cytoskeleton was affected in FOSB-transfected A549 cells but not in nontransfected or EGFP-transfected cells (Appendix A). Approximately 40% of FOSB-expressing cells exhibited a collapsed microtubule network (Appendix A). Interestingly, TP4 disrupted microtubules (Appendix A and Ting et al.) with ~52% of cells showing a collapsed microtubule work (Appendix A). We then asked whether FOSB knockdown can ameliorate TP4-caused microtubule defects. The results showed that microtubule collapse events were partially prevented in FOSB-knockdown cells, in which only 20.6% of TP4-treated cells exhibited collapsed microtubules (Appendix A). These results supported the notion that FOSB signaling causes cytoskeletal defects which is independent from microtubule disruption caused by TP4. We measured the levels of a panel of epithelial-to-mesenchymal transition (EMT) and cytoskeletal proteins in A549 cells with FOSB-overexpression or TP4 treatment. Among the proteins we examined, E-Cadherin, N-Cadherin, Integrin-α5, and Stathmin levels were decreased, while PCDHB13 was increased upon FOSB overexpression or TP4 treatment compared to respective controls (Figure 2C–F). Vimentin and αSMA levels were not significantly affected by either treatment (Figure 2D,F). Knockdown of FOSB prevented effects of TP4, with no significant differences in E-Cadherin, N-Cadherin, or PCDHB13 levels (Figure 2G,H). These findings suggested that TP4 caused FOSB-dependent dysregulation of cell matrix proteins. Notably, we only observed upregulation of PCDHB13 by TP4 in NSCLC cell lines but not normal cells (Figure 2E and Appendix A), suggesting that PCDHB13 plays a specific role in NSCLC.

### 2.3. PCDHB13 as a Downstream Target of FOSB

To further characterize the role of PCDHB13 in NSCLC, we investigated whether PCDHB13 is a downstream target of FOSB. We examined its protein level in TP4-treated FOSB-knockdown cells and found it was diminished compared to controls (Figure 3A,B). We then performed a bioinformatics analysis of the PCDHB13 promoter sequence with Tfsitescan [36] and identified one FOS promoter basal sequence and multiple putative binding elements (PBE1: −771 to −777; PBE2: −921 to −927; and PBE3: −1019 to −1025), which are expected to be recognized by FOSB (Figure 3C). Three different primer sets were designed for chromatin immunoprecipitation (ChIP)-qPCR to validate FOSB targets in the promoter (Figure 3C), and the assay confirmed that FOSB targets the PBE1-PBE3 regions (Figure 3D–G). In addition, the PCDHB13 promoter-driven mCherry reporter was transfected alone or along with FOSB-tGFP in A549 cells. At 48 h post-transfection, a fluorescence signal was observed (Figure 3H) and cell lysates were probed for mCherry and tGFP. mCherry fluorescence was increased by forced expression of FOSB (Figure 3I,J), suggesting that FOSB modulates PCDHB13 promoter activity.

### 2.4. PCDHB13 Affects Microtubule Dynamics

To further characterize the function of PCDHB13, we investigated its cellular localization in PCDHB13-transfected cells. The predicted membrane localization pattern was not observed, but instead we observed a cytosolic filamentary structure that was partially associated with microtubules (Figure 4A). The three-dimensional orientation of the PCDHB13–microtubule network was analyzed by Imaris software (Figure 4B and Appendix A), revealing irregular grid-like patterns in the distal cytoplasm and dendritic filaments in the proximal cytoplasm (Figure 4B). Co-immunoprecipitation (Co-IP) was then performed, confirming that PCDHB13 interacts with microtubules (Figure 4C). To investigate the role of PCDHB13 in microtubule regulation, we used a microtubule regrowth assay. Cells were treated with nocodazole to destabilize microtubule networks; after nocodazole washout, ICC was conducted to analyze the formation of microtubule organizing centers (MTOCs) and microtubule polymerization. The formation of MTOCs was observed immediately (time point 0′) after washout in both control and PCDHB13-overexpressing cells (Figure 4D,E). Most MTOCs formed within 30 min, indicating that MTOC formation was not affected (Figure 4F). In the control group, polymerized microtubules emerged rapidly forming complex network structures within 30 min (Figure 4D). However, PCDHB13-overexpressing cells exhibited defective microtubule outgrowth patterns at 30 min (Figure 4E). To further examine the spatial association of overexpressed PCDHB13 with microtubules, colocalization was quantified by a line-series intensity correlation. PCDHB13 showed similar localization pattern to microtubules with some cytosolic aggregates observed at 10 and 30 min (Figure 4E). A proposed model for the PCDHB13-driven microtubule abnormality was shown in Figure 4G. These findings suggested that PCDHB13 overexpression disrupted microtubule organization, so we further asked which domain of PCDHB13 is required to disrupt microtubule structure. Both the short form (aa22–353) and long form (aa29–690) of the putative extracellular cadherin (EC) domain strongly ‘enhanced’ microtubule polymerization, but the cytoplasmic region (CR) did not (Figure 5A–D). Notably, the fluorescence intensity in EC domain-expressing cells was higher than controls immediately upon incubation with tubulins, suggesting a rapid association between the EC domain and tubulins (Figure 5B,C). These findings seemed to conflict with the observation that elevated expression of PCDHB13 caused cellular defects in microtubule polymerization; however, the presence of PCDHB13–microtubule-containing aggregates (Figure 5E) raised the possibility that PCDHB13 may trigger tubulin polymerization abnormalities. To test this possibility, fluorescent dye-conjugated tubulins were incubated with the EC domains and observed by microscopy. In controls, tubulins were highly polymerized upon paclitaxel treatment and depolymerized by nocodazole (Figure 5(Ei,ii)), but irregularly polymerized microtubule aggregates were observed upon incubation with EC domains (Figure 5(Eiii–vi)), indicating that cytosolic PCDHB13 affects microtubule structure.

### 2.5. Suppressive effects of PCDHB13 in NSCLC

Since PCDHB13 was found to be associated with microtubules, we asked whether expression of PCDHB13 may affect NSCLC cell properties. A549 cells were seeded on Matrigel-coated transwell inserts, and cell invasion was observed 24 h later. Invasion was significantly decreased in cells transfected with PCDHB13 or FOSB compared to GFP-transfected cells (Figure 6A,B). In addition, relatively nontoxic doses of TP4 also inhibited cell invasion (Figure 6C,D). To investigate the role of the FOSB–PCDHB13 axis in vivo, PCDHB13- or FOSB-transfected cells were xenotransplanted via retrol-orbital (RO) injection into transparent zebrafish (mitfa^b692^) that lack pigment cells (Figure 6E). Four days after xenotransplantation, internal organs were eviscerated for microscopic and Western blot analysis. A lower fluorescence signal was observed in fish receiving PCDHB13- or FOSB-transfected cells compared to those receiving tGFP-transfected controls (Figure 6F,G). Xenotransplanted cells were detected by N-Cadherin antibody (Figure 6H). We observed that both PCDHB13 and FOSB protein levels (reflected by tGFP level) were decreased compared to tGFP (Figure 6H,I), indicating that overexpression of PCDHB13 or FOSB suppressed NSCLC cell proliferation. Since we observed FOSB overexpression induces cell death (Figure 1B), we asked whether overexpression of PCDHB13 may be cytotoxic. Indeed, PCDHB13 overexpression in NSCLC cells reduced viability (Figure 6J). TP4 dominantly causes cellular necrosis [14,17], so we next assessed whether overexpression of PCDHB13 and FOSB may also induce necrosis in NSCLC cells. The result showed that both PCDHB13 and FOSB overexpression caused LDH release, suggesting that necrosis had occurred (Figure 6K). We then evaluated whether TP4 caused toxicity is dependent on PCDHB13. PCDHB13 expression was knocked down by siRNA in A549 cells (Figure 6L), and cell viability was determined after TP4 treatment (Figure 6M). Interestingly, PCDHB13 knockdown did not completely block TP4 caused cell death (Figure 6M), suggesting that PCDHB13 activation may be one of multiple events that occurs after TP4 treatment.

### 2.6. Dysregulation of PCDHB13 and FOSB in NSCLC

Low expression of FOSB and PCDHB13 in NSCLC suggests a suppressive role in tumorigenesis, so we assessed the prognostic value of these two factors using an online interface to query a transcriptomic database [37]. Higher expression levels of both genes in NSCLC were correlated with improved survival status (Appendix A). We then evaluated protein levels in lung cancer tissue arrays by IHC, finding that both proteins were clearly detectable in matched normal adjacent tissue (NAT) but not in tumor samples (Figure 7A and Appendix A). Moreover, protein signals were significantly higher in tumors from the surviving group compared to the nonsurviving group (Figure 7B,C, *p* < 0.001). Furthermore, the level of PCDHB13 was positively correlated with the level of FOSB (*r* = 0.6070, *p* < 0.001). To test whether protein expression correlates with mortality, we quantified the levels of FOSB and PCDHB13 in each tumor sample and normalized the values to the average from nine matched NAT samples. A normalized value over 1.0 was defined as high expression, while a value less than 1.0 indicated low expression. Both proteins showed a trend toward decreased expression with advancing NSCLC stages (Figure 7E,F). In addition, the group with high tumor expression of FOSB (*n* = 26) and PCDHB13 (*n* = 6) had significantly better survival status than the low expression group (Figure 7G, log-rank test, *p* < 0.001; *p* < 0.05). To validate the levels of FOSB and PCDHB13 in NSCLC, tissue samples from twelve patients with recorded diagnosis were used for immunoblotting (Figure 7H–J and Appendix A). FOSB or PCDHB3 expression was detected in the NAT from three patients (No. 1, 5, and 6), but was downregulated in eleven out of twelve patients. These findings confirmed that PCDHB13 and FOSB were suppressed in NSCLC and suggest that these two proteins might be associated with NSCLC development.

## 3. Discussion

Here we show mitochondrial-stress induced FOSB activation by TP4 is required Ca^2+^ elevations in NSCLC cells. FOSB causes EMT-independent cytoskeletal changes and upregulation of PCDHB13, which was associated with microtubules, indicating a distinctive role unrelated to cell–cell adhesion. These results suggest that dysregulation of the FOSB–PCDHB13 axis may contribute to phenotypic changes in NSCLC cells and the proteins may serve as useful biomarkers of disease progression.

Distinct compositions of AP-1 dimers have been shown to influence cancer status. As such, constitutive activation of cJUN is frequently observed in cancer, and dominant-negative cJUN has been reported to inhibit cell growth [38,39]. In contrast to the JUN family, FOS proteins appear to play more diverse roles in cancer. Gene profiling studies suggested that FOSB-regulated *MMP9* is a potential target for preventing NSCLC development [40]. However, both c-FOS and FOSB were shown to be downregulated in NSCLC [23], suggesting a tumor suppressive role. In contrast, FRA1 and PREP-1 are required for EMT and metastasis in NSCLC [41]. FRA1 was shown to potently inhibit cellular differentiation [42], but it is insufficient to promote tumor formation [43]. In TNBC, FRA1 depletion suppressed tumor proliferative and invasive phenotypes [26]. We have shown that strong induction of FOSB by TP4 disrupts FRA1/AP-1-regulated transcription and EMT [14]. However, TP4 treatment of NSCLC cells did not alter FRA1-associated protein levels (Appendix A), suggesting that FRA1-associated pathways are differentially regulated between NSCLC and TNBC. Transcriptomic studies on FOSB-overexpressing cells suggests that induction of FOSB may dysregulate NSCLC progression by promoting cytoskeletal and morphological changes (Figure 2A,B and Appendix A), which affects NSCLC fate. However, the mechanisms by which potential competition between FRA1 and FOSB influence the NSCLC transcriptome remain to be further elucidated.

One major function of *PCDH* genes is thought to be in neuronal specification [44,45]. However, like classical cadherins, protocadherins contain putative ECs that are capable of mediating adhesion of multiple cell types and distinct CRs that can transduce intracellular signaling in various contexts [45]. The CRs of clustered PCDHs do not include the catenin-binding domain [45], and the CRs of PCDHAs and PCDHGs differ from those of PCDHBs. The CRs of PCDHA/G are encoded by conserved exons and are required for interactions with RET [46], while the role in PCDHBs has not been characterized. Interestingly, we observed that PCDHB13 interacts with microtubules (Figure 4C) through the EC domain (Figure 6), indicating that PCDHB13 may regulate microtubule dynamics. The microtubule network is required for biogenesis of cadherin-associated vesicles and coordinately regulates Cadherin-mediated cell adhesion [46]. We observed that TP4 treatment or FOSB overexpression decreased classical Cadherins (Figure 2C–F), suggesting that cadherin-mediated microtubule regulation was affected and these cellular changes are EMT-independent. In addition, stathmin was downregulated upon TP4 treatment or FOSB overexpression in A549 cells (Figure 2C–F). Stathmin promotes microtubule disassembly, regulating MTOC polarization [47], and as such, decreased stathmin levels may promote microtubule dysregulation in NSCLC. Furthermore, PCDHB13 is probably not involved in MTOC formation (Figure 4E,F), but its overexpression does cause microtubule abnormalities (Figure 4D,E). The EC domains of PCDHB13 promote irregular tubulin polymerization (Figure 6), indicating that PCDHB13 affects microtubule dynamics. Moreover, FOSB-knockdown cells only showed partial recovery of microtubule defects caused by TP4 treatment (Appendix A), suggesting a FOSB/PCDHB13-independent role of TP4 on microtubule disruption. Indeed, we have observed that penetrated TP4 targets to the microtubule and leads to the microtubule depolymerization [15]. Overall, we propose that EMT-independent transcriptional reprogramming is mediated by FOSB, which consequently influences microtubule dynamics and causes cell death.

## 4. Materials and Methods

### 4.1. Ethics Approval

All fish experimental procedures were in accordance with Academia Sinica guidelines and were approved by the Ethical Committee for Using Vertebrates as Experimental Animals. The use of human tissue specimens from the NSCLC patients was approved by the Institutional Review Board (IRB) at Shuang Ho Hospital, Taipei Medical University (IRB No: N201702026). Commercially available human tissue samples were used according to the regulations set by the Human Subject Research Ethics Committee of Academia Sinica.

### 4.2. Reagents and Plasmids

TP4 (FIHHIIGGLFSAGKAIHRLIRRRRR) and N-terminal-biotinylated TP4 were synthesized and purified by GL Biochem Ltd. (Shanghai, China) as previously described [14]. 1,2-Bis(2-aminophenoxy)ethane-N,N,N’,N’-tetra acetic acid tetra kis(acetoxymethyl ester) (BAPTA/AM) and Ethyl 3-aminobenzoate methane sulfonate (MS-222) were purchased from Sigma-Aldrich (St. Louis, MO, USA). N-(4-Amino-5-cyano-6-ethoxypyridin-2-yl)-2-(2,5-dimethoxyphenyl)acetamide (JNK inhibitor VIII) and 2-(2-Amino-3-methoxyphenyl)-4H-1-benzopyran-4-one (PD98059) were purchased from Calbiochem (Merck Millipore, Temecula, CA, USA). Recombinant PCDHB13 was purchased from NovoPro Bioscience, Inc. (Shanghai, China) (amino acids 22–353) and MyBiosource, Inc. (San Diego, CA, USA) (amino acids 29–690). The *PCDHB13* promoter-driven mCherry reporter construct, tGFP-tagged FOSB/PCDHB13, monomeric GFP (mGFP)-tagged FOSΔB, and pCMV6-ac-GFP (tGFP) were purchased from GeneCopoeia, Inc. (Rockville, MD, USA). The pEGFP-C1 vector is from Clontech Laboratories (Foster City, CA, USA). Primary antibodies were purchased from the Cell signaling (Boston, MA, USA) (FOSB, clone 5G4; FRA1,clone D80B; JUNB, clone C37F9; N-Cadherin; E-Cadherin, clone 24E10; Integrin α5; ERK1/2; phospho-ERK1/2; SAPK/JNK; phospho-SAPK/JNK; Stathmin; α-Tubulin, clone DM1A; α/β-Tubulin; MMP-2, clone D4M2N), EMD Millipore (Calreticulin; JUND; c-JUN, clone 6A6.2; GAPDH, clone 6C5), OriGene Technologies (Rockville, MD, USA) (Vimentin; αActin (smooth muscle); PCDHB13), ABGENT (San Diego, CA, USA) (SNAIL, clone RB14258), Abcam (Cambridge, MA, USA) (Giantin), Santa Cruz Biotechnology (Santa Cruz, CA, USA) (Biotin clone 39-15D9; Prohibitin), and Cell BioLabs (San Diego, CA, USA) (αRFP, clone RF5R). Alexa Flour fluorescent dye-conjugated secondary antibodies were purchased from Molecular Probes (Carlsbad, CA, USA). Generation of the TP4 antibody was commissioned by MDBio, Inc. (Taipei, Taiwan) as previously described [15].

### 4.3. Cell Culture and Knockdown Assay

Cell lines were purchased from the Bioresource Collection and Research Center ((BCRC, Hsinchu, Taiwan, including A549 (BCRC) 60074), NCI-H661 (BCRC 60125), NCI-H209 (BCRC 60123), and MRC-5 (BCRC 60023)) as well as the American Type Culture Collection (ATCC, VA, USA, including NCI-H1975 (ATCC CRL-5908) and BEAS-2B (ATCC CRL-9609)) and cultured as recommended. For the cell viability and transfection assay, 5 × 10^3^ cells were seeded into the wells of a 96-well plate and cultured overnight. For the transfection assays, cells were transfected with 0.1–0.4 μg FOSB/FOSΔB/PCDHB13 expression plasmids and cell viability was determined after 72 h. During the drug treatment assay, inhibitors were added 30 min prior to TP4, and cell viability was determined at indicated time points. Transfection was performed using LipofectAMINE^TM^3000 (ThermoFisher Scientific, Carlsbad, CA, USA) as previously described [13]. Knockdown was performed in A549 cells with Silencer® Select presynthesized siRNAs (ThermoFisher Scientific) targeted to the *FOSB* (ID: s223612, s230577) and *PCDHB13* (ID: s31938, s31939) using Lipofectamine^®^ RNAiMAX Reagent (ThermoFisher Scientific) in accordance with the manufacturer’s protocol. Nontargeting siRNA (Negative Control siRNA#1, ThermoFisher Scientific) was used as a control for the knockdown assay.

### 4.4. Cell Viability Assay

Cell viability was quantitatively analyzed using the CellTiter-Glo^®^ Luminescent Cell Viability Assay kit (ATP assay) (Promega, Madison, WI, USA) and Lactate dehydrogenase (LDH) release was quantified with a Cytotoxicity Detection KitPLUS (LDH) (Roche Applied Science, Basel, Switzerland) as previously described [14]. The luminescent signal was measured with a photometer (SpectraMax^®^ i3, Molecular Devices, Lagerhausstrasse, Wals, Austria).

### 4.5. Transcriptome Analysis

Total RNA samples were extracted from FOSB-transfected or control GFP-transfected A549 cells at 40 h post-transfection. Sample preparation procedures and data analysis for microarray studies were performed as previously described [14]. Transcriptomic data were deposited on the Gene Expression Omnibus (GEO) database (GEO accession number: GSE111827).

### 4.6. Chromatin Immunoprecipitation and Quantitative-PCR

Chromatin immunoprecipitation (ChIP) assays were performed using the Pierce™ Magnetic ChIP Kit (Thermo Fisher Scientific, Rockford, IL, USA) according to the manufacturer’s protocol. The primary antibodies used in IP reactions were anti-RNA polymerase II (positive control), Rabbit IgG antibody (negative control), and anti-FOSB antibody. Quantitative PCR was performed by using the QuantiNova SYBR Green PCR kit (Qiagen, GmbH, Hilden, Germany), following the recommended protocol. The PCR conditions were 95 °C for 2 min (heat activation) and 40 cycles of 2-step cycling (95 °C/5 s, 60 °C/25s for P1-P3, and 10 s for GAPDH). After PCR, a melting curve was generated. Fold enrichment of target-specific ChIP signal was calculated relative to the isotype antibody (IgG) control using the formula: fold enrichment = 2^^Raw [The threshold cycle number (C^_t_^)]^, where Raw C_t_ = C_t__−IP_ − C_t__−IgG_. Primer pairs were designed and validated by using MacVector software (MacVector, Inc., Cary, NC, USA). Primer sequences for the qPCR were as follows: P1F:5′-GGATCTTGATTGAAACAACAGC-3′; P1R:5′-GGATGTCCTTTTTTATTACCTCCAC-3′; P2F:5′-GGGATGTGAAAATGTTGAAGGG-3′; P2R:5′-GGAAATGAACAAGGAAAGGC-3′; P3F:5′-CTCACTTTACAGAGGAGCAAAC-3′; P3R:5′-TCCCAACTCTGCCTGTATG-3′. The predicted amplicon sizes were 381 (P1), 264 (P2), and 325 (P3) bp. DNA agarose (1.5%) electrophoresis was conducted to confirm PCR specificity. The putative FOS binding elements in the *PCDHB13* promoter (Entrez_ID = 56123; Genome = hg38; chr5+:141,212,566–141,214,103 bp) were predicted by the Tfsitescan program (http://www.ifti.org/Tfsitescan/) (Institute for Transcriptional Informatics, Pittsburgh, PA, USA).

### 4.7. Tissue Fractionation, Co-Immunoprecipitation, and Western Blot

Mitochondrial isolation was performed using the Mitochondria Isolation Kit for Culture Cells (Thermo Fisher Scientific, Rockford, IL, USA) following the suggested protocol. Membrane fractionation was performed using the Mem-PER™ Plus Membrane Protein Extraction Kit (Thermo Fisher Scientific) according to the suggested protocol. Cells were treated with 5.03 μM TP4 in membrane fractionation experiments. For the co-immunoprecipitation (coIP) experiment, equal amounts of protein lysate (500 μg) form PCDHB13-tGFP- and control tGFP-overexpressing A549 cells were harvested for IP using anti-tGFP magnetic beads (OriGene Technologies), in accordance with the recommended protocol. Cell extract preparation and Western blot were performed as previously described [14]. Protein samples were loaded in duplicate gels and transferred to separate membranes for probing with different antibodies. The results were expressed as relative densitometric units (RDU; densitometric intensities of FOSB+FOSΔB divided by that of GAPDH). Complete and unedited Western blot images were shown in Appendix A.

### 4.8. Calcium Measurement

Calcium (Ca^2+^) levels were determined using the Fluo-4 Direct Ca^2+^ assay kit (ThermoFisher Scientific) and Rhod-2 calcium indicator (ThermoFisher Scientific), as previously described [14].

### 4.9. Mitochondrial Activity Assay

Mitochondria were stained by MitoTracker™ Red CMXRos dye (ThermoFisher Scientific) according to manufacturer instructions. TP4 was treated at doses 5.03-6.71 μM for 3 h. Cells were fixed with 4% PFA, mounted by ProLong Anti-fade mounting medium (Thermo Fisher Scientific), and observed under a confocal microscope. The fluorescence signal (which is proportional to mitochondrial membrane polarization) was quantitatively determined using Image J software.

### 4.10. Reporter Analysis

For the *PCDHB13* promoter reporter assay, the p*PCDHB13*-mCherry vector (0.5 μg) was cotransfected with FOSB-tGFP (0.5, 1.0, and 2.0 μg) in 2 × 10^5^ A549 cells. Cells were harvested 48 h post-transfection for microscopic and Western blot analyses. Images were captured with an inverted fluorescence microscope equipped with a digital camera (Olympus IX71/DP80, Tokyo, Japan). CellSens software was used for image acquisition.

### 4.11. Immunocytochemical and Immunohistochemical Studies

For confocal microscopic analysis, antibodies stained samples were mounted with fluorescence mounting medium (ProLong Gold Antifade Reagent, ThermoFisher Scientific, OR, USA) and images were obtained with a FV1000 or FV3000 laser-scanning confocal microscope (Olympus, Tokyo, Japan). ASW2.1 and FV31S software (Olympus) were used for image acquisition. The spatial colocalization pattern and relative fluorescence intensities of PCDHB13 and α-Tubulin were analyzed using the line-series analysis in ASW2.1 software. The three-dimensional colocalization of PCDHB13 and α-Tubulin was determined using Imaris software (v.9.0.0, Bitplane, Zurich, Switzerland). The fluorescence signal was quantified using Image J software. Human lung carcinoma tissue arrays were purchased from SUPER BIO CHIPS (CC5 and CCA4) (Seoul, Korea). Paraffin sections were processed for immunostaining and Hoechst33342 or 3,3’-diaminobenzidine tetrahydrochloride (DAB) staining using the Ultra-sensitive ABC peroxidase kit and Pierce™ DAB Substrate Kit (Thermo Fisher Scientific) by standard protocols. Images were obtained with an inverted microscope and quantified with Image J software (National Institutes of Health, USA) using the IHC tool box plugins.

### 4.12. Microtubule Regrowth and In Vitro Tubulin Polymerization Assay

The in vivo tubulin polymerization assay was performed according to the “enhancer condition” in the manufacturer’s instructions (Cytoskeleton, Inc., Denver, CO, USA). The microtubule regrowth assay was conducted as described previously [15].

### 4.13. Cell Invasion Assay

The cell invasion assay was performed following the manufacturer’s instructions (Corning, Inc., Corning, NY, USA). Before the assay, the transwell insert (polycarbonate membrane with 8 μm pore size, Corning, Inc.) was coated with 100 μL diluted Matrigel (1–5 mg mL^−1^, Corning, Inc.) overnight in 5% CO_2_ incubator at 37 °C to promote gelling. In addition, A549 cells were seeded on a 10-cm plate and grown to 95% confluence. At 24 h post-transfection (EGFP, PCDHB13-tGFP, FOSB-tGFP), cells were trypsinized and washed three times with Ham′s F12K medium (Thermo Fisher Scientific), containing 1% FBS. 1 × 10^6^ cells were resuspended in culture media containing 1% FBS. Before seeding 100 μL transfected cells in the upper chamber, gelled Matrigel was washed by prewarmed serum-free culture medium. The lower chamber of the transwell was filled with 600 μL of culture medium containing EGF (5 μg mL^−1^, Thermo Fisher Scientific). After 24 h incubation, inserts were removed and fixed by 4% PFA for 15 min at RT and stained by Hoechst33342. Non-invading cells that remained on the top of the transwell were scraped by a cotton swab. Transwell inserts were placed on a glass slide and cells that had invaded were observed by an inverted fluorescent microscope equipped with a digital camera.

### 4.14. Zebrafish Xenotransplantation Model

*Mitfa*^b692^ line zebrafish (*Danio rerio*) were provided by the Taiwan Zebrafish Core Facility (Taipei, Taiwan) and were cultured in a 14:10 h light–dark cycle at 28 °C. Tumor cell xenotransplantation was performed through RO injection, following previously published methods with modifications [48]. Before cell transplantation, A549 cells were seeded on a 10-cm plate and grown to 95% confluence. At 24 h post-transfection, cells were trypsinized and washed twice with PBS containing 5% FBS and 1% Pen-Strep. Cells were resuspended in 200 μL of PBS containing 5% FBS and 1% Pen-Strep and passed through a round bottom polystyrene tube with 35-μm strainer. A total 3–4 μL of cell suspensions (~1 × 10^5^ cells) were injected using a 10-μL Hamilton syringe with a removable 26-gauge 15-mm length needle. Before injection, five-month-old adult *mitfa*^b692^ strain zebrafish were anesthetized in MS-222 (4 mg mL^−1^) for 45 s. Fish were placed on a sponge and cells were injected at the seven o’clock position with 45° angle into the fish eye. Xenotransplanted fish were cultured with aeration at 30 °C for 4 days. Internal organs were eviscerated and harvested for fluorescent microscopic analysis and Western blot. Images were taken with a fluorescent microscope that was equipped with a digital camera, using a 2× objective lens (Olympus PLAPON 2×, N.A. 0.08). The fluorescent signal was quantitatively determined using Image J software.

### 4.15. Statistical Analysis

For multiwell-based assays, cells were plated in at least quadruplicate. Data were collected from independently repeated experiments (*n* ≥ 3) and were analyzed by Prism 5 software (GraphPad Inc., San Diego, CA, USA). A two-tailed Student’s *t*-test or one-way/two-way ANOVA with Bonferroni post-hoc test was used to determine statistical significance.

## 5. Conclusions

We conclude that mitochondrial stress-induced Ca^2+^ signaling regulated FOSB–PCDHB13 axis is harmful to the NSCLC development. These findings introduce FOSB–PCDHB13 axis as a therapeutic target for NSCLC.

## Figures and Tables

**Figure 1 cancers-11-00107-f001:**
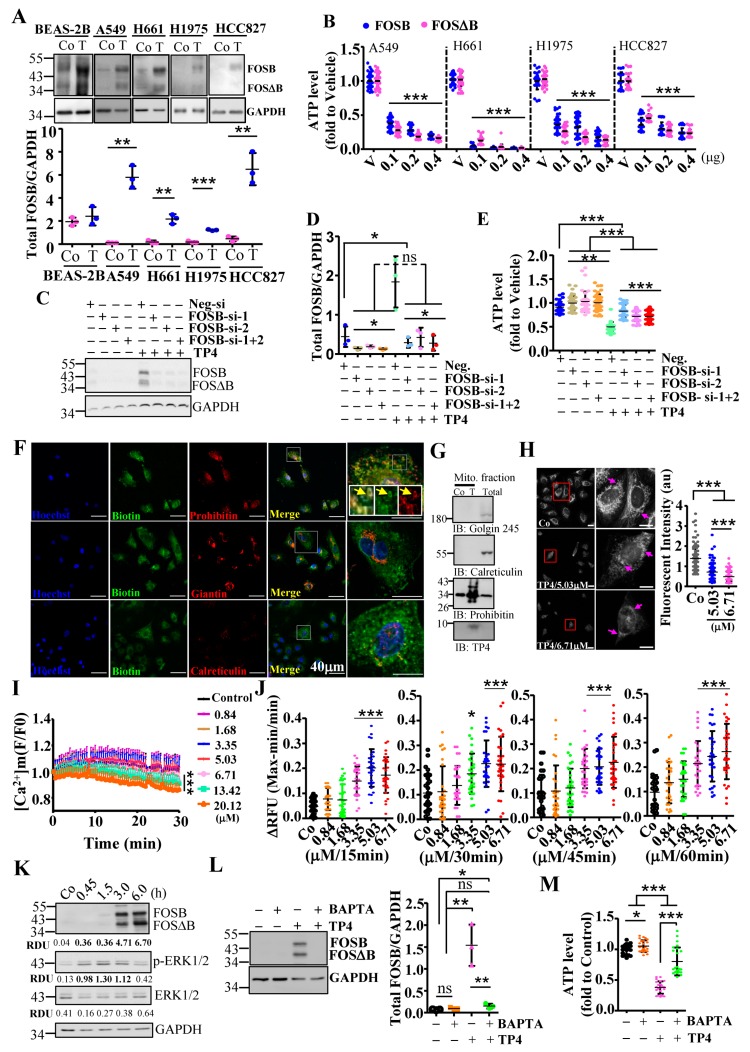
Mitochondrial stress-mediated FOSB activation in non-small cell lung cancer (NSCLC) cells. (**A**) Total lysates from control and NSCLC cells without (Co) or with TP4 treatment (T, 6.71 μM) were analyzed by Western blot using antibodies against GAPDH and FOSB. Quantitative analysis of FOSB (normalized to GAPDH) is shown below the blot. (**B**) Cell viability was determined by the ATP assay for NSCLC cells transfected with FOSB or FOSΔB plasmid. Eight replicate wells were analyzed for each dose. (**C**) Total lysates from A549 cells transfected with control (Neg-si) or FOSB small interfering RNAs (siRNAs) were analyzed by Western blot using antibodies against GAPDH and FOSB. (**D**) Quantitative analyses of FOSB levels (**C**), normalized to GAPDH. (**E**) Viability of A549 cells transfected with FOSB siRNAs were determined by ATP assay. Eight replicate wells were analyzed for each condition. (**F**) Intracellular localization of biotinylated-TP4 in A549 cells. Cells were stained with biotin, prohibitin (upper), giantin (middle), and calreticulin (lower) antibodies. Hoechst33342 was used to stain nuclei. Boxed regions are magnified in the panels to the right of the merged images. Yellow arrows indicate colocalization of biotinylated-TP4 with mitochondria. Bar: 40 μm. (**G**) Mitochondrial fractions from A549 cells without (Co) or with TP4 treatment (T, 5.03 μM) were analyzed by Western blot using antibodies against organelle markers and TP4. (**H**) Mitochondria in A549 cells were stained by MitoTracker Red CMXRos dye. Fluorescence intensity of the mitochondria in each cell was quantified after 5.03 and 6.71 μM TP4 treatment for 3 h. Bar: 20 μm. (**I**) Mitochondrial Ca^2+^ levels were measured kinetically (every 30 s for 30 min) using Rhod-2 AM dye after treatment with the indicated doses of TP4. (**J**) Ca^2+^ levels were measured by the addition of Fluo-4 dye after treatment with the indicated doses of TP4 for 15–60 min. Eight wells were analyzed for each treatment in an independent repeat. (**K**) Total lysates from A594 cells without (Co) or with 6.71 μM TP4 treatment (T) for 45 min to 6 h were analyzed by Western blot using antibody against GAPDH, FOSB, and ERK. The relative amounts of FOSB + FOSΔB, ERK1/2, and p-ERK1/2 in each lane are expressed as RDU and normalized to GAPDH signal. Experiments were independently repeated with comparable results. (**L**) Total lysates from control, BAPTA/AM-treated, TP4-treated cells, and combination-treated cells were analyzed by Western blot using antibodies against GAPDH and FOSB. Quantitative analyses of the blots shown in left; levels of FOSB + FOSΔB were normalized to GAPDH. (**N**) Cell viability was measured in cells treated with BATPA/AM and TP4. Eight wells were analyzed for each independent replicate. Quantitative results are presented as the mean ± SD. (*n* = 3, two tailed *t*-test in (**A**,**D**,**E**,**H**,**J**,**L**,**M**), one-way ANOVA followed by Bonferroni’s test in (**B**): * *p* < 0.05; ** *p* < 0.01; *** *p* < 0.001). Co in (**A**,**G**,**K**): control group.

**Figure 2 cancers-11-00107-f002:**
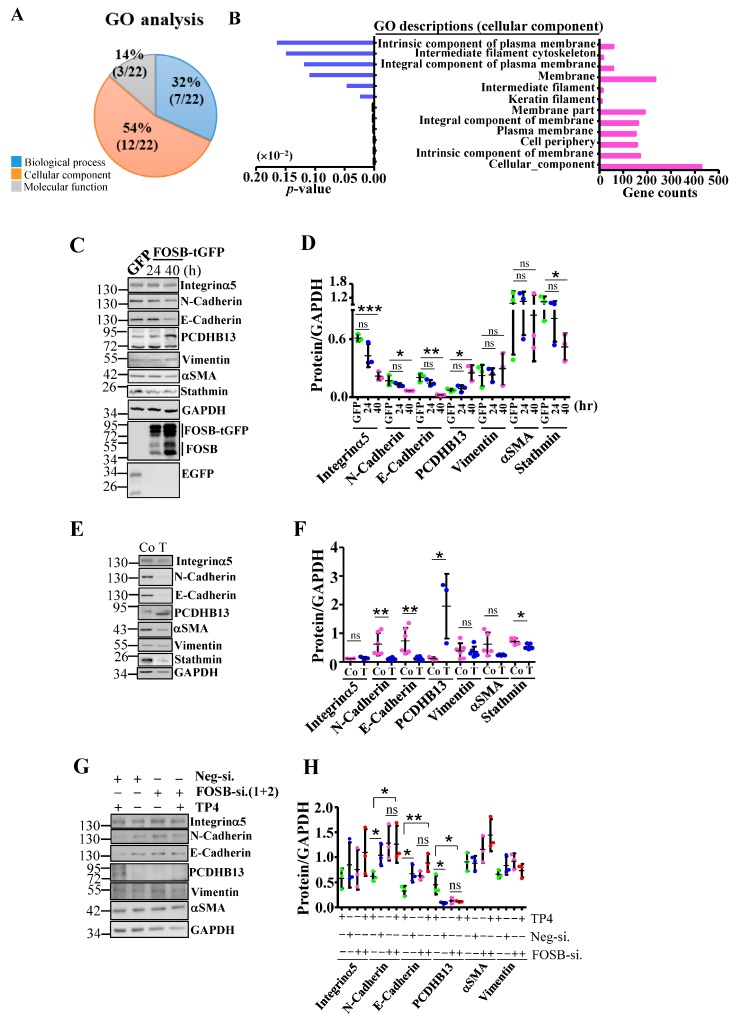
Loss of cytoskeletal integrity upon FOSB induction. (**A**,**B**) Gene ontology (GO) analyses of dysregulated genes revealed three distinct functional categories (**A**). Twelve out of twenty-two annotation terms were assigned to the cellular component ontology, including genes that are involved in the regulation of cytoskeleton and membrane (**B**). (**C**,**D**) Total lysates from A549 cells transfected with EGFP or FOSB-tGFP plasmid were analyzed by Western blot using antibodies against GFP, GAPDH, FOSB, EMT markers, PCDHB13, and Stathmin. (**E**,**F**) Total lysates from A549 cells without (Co.) or with TP4 (T) were analyzed by Western blot using antibody against GAPDH, EMT markers, PCDHB13, and Stathmin. (**G**,**H**) Total lysates from A549 cells transfected with control (Neg-si) or FOSB siRNAs (FOSB-si-1+2) with or without TP4 treatment were analyzed by Western blot using antibodies against GAPDH, EMT markers, and PCDHB13. Quantitative measurements of protein levels were normalized to GAPDH in (**D**,**F**,**H**). Quantitative results represent the mean ± SD (*n* = 3, two-tailed *t*-test in (**D**,**G**,**H**): * *p* < 0.05; ** *p* < 0.01; *** *p* < 0.001).

**Figure 3 cancers-11-00107-f003:**
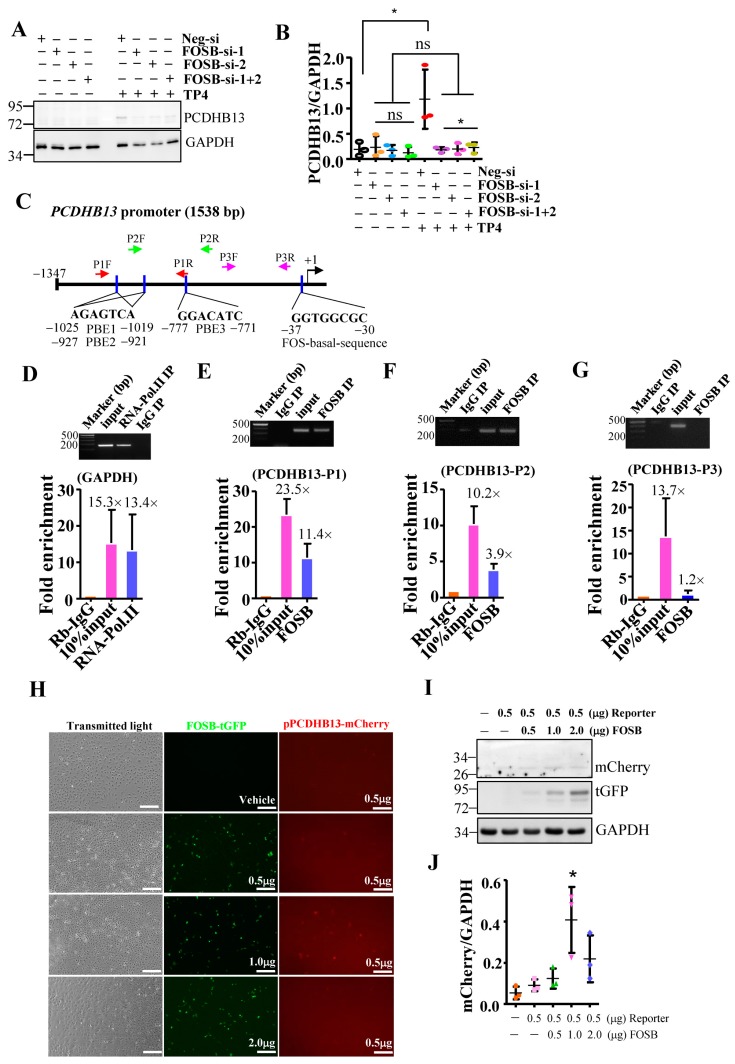
PCDHB13 is a downstream target of FOSB. (**A**) Total lysate from A549 cells transfected with control (Neg.) or FOSB siRNAs were analyzed by Western blot using antibodies against GAPDH and PCDHB13. (**B**) Quantitative analyses of PCDHB13 levels in (**A**) normalized to GAPDH. (**C**–**G**) Quantitative PCR following ChIP assay to confirm putative FOSB target sites in the PCDHB13 promoter. Two primer sets were used for qPCR studies (**C**). RNA Pol II antibody and a qPCR primer-pair flanking the GAPDH promoter served as positive controls, while non-specific rabbit IgG antibody served as a negative control. The ChIP-qPCR signals are normalized to Rabbit IgG signals, showing the fold enrichment relative to the background (**D**–**G**). Results represent mean ± SD. PCR amplicons were electrophoresed on a 1.5% agarose gel to confirm specificity of amplification. (**H**) Transmitted light and fluorescent images of A549 cells transfected with reporter or cotransfected with reporter and FOSB. Bar: 100 μm. (**I**) Total lysates from A549 cells transfected with reporter or cotransfected with reporter and FOSB were analyzed by Western blot using antibodies against tGFP, GAPDH, and FOSB. (**J**) Quantitative measurements of protein levels were normalized to GAPDH (**I**). Quantitative results represent the mean ± SD (*n* = 3, two tailed *t*-test: * *p* < 0.05).

**Figure 4 cancers-11-00107-f004:**
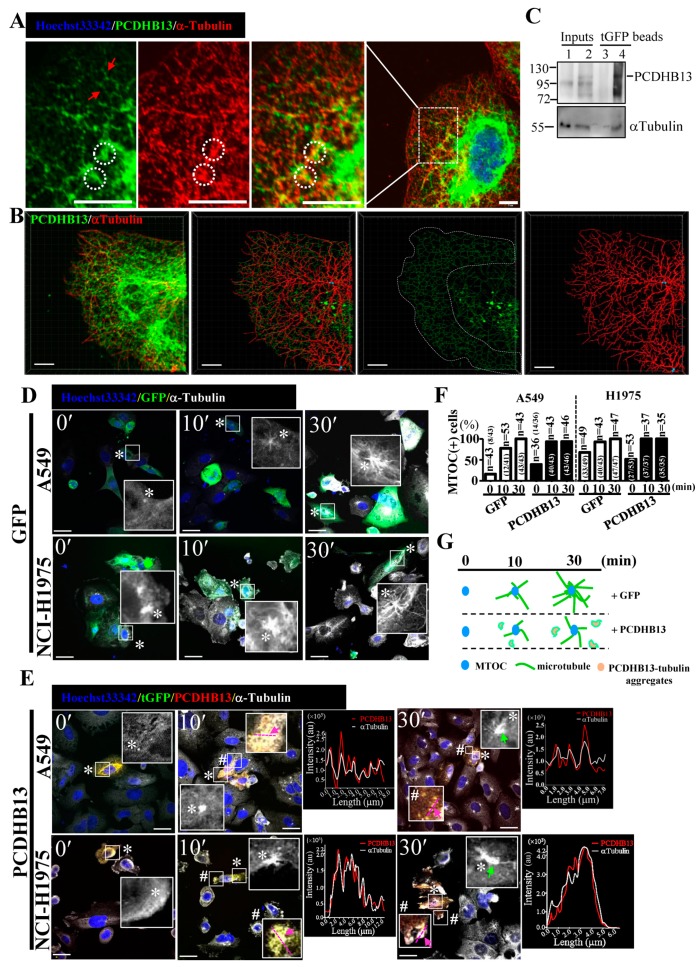
PCDHB13 disrupts microtubule dynamics in NSCLC cells. (**A**) Cellular localization of PCDHB13 in A549 cells. Cells were stained with PCDHB13 (green) and α-Tubulin (red). Hoechst33342 was used to stain nuclei (blue). Boxed regions are magnified in the panels to the right. Red arrows indicate the filamentary structure of PCDHB13. White circles indicate the cellular aggregates formed by PCDHB13 and α-Tubulin. Bar: 20 μm. (**B**) PCDHB13 and α-Tubulin are shown in red and green, respectively. Cytoskeletal projections of each protein were simulated by Imaris software. Yellow color indicates colocalization of PCDHB13 and α-Tubulin. Bar: 8 μm. (**C**) A549 cell lysates without (lane 1 and 3) or with tGFP-tagged PCDHB13 overexpression (lane 2 and 4) were harvested for co-immunoprecipitation (Co-IP) assay using the anti-tGFP conjugated magnetic beads. IP eluates were analyzed by Western blot for PCDHB13 and α-Tubulin. (**D**,**E**) GFP (**D**) or PCDHB13-tGFP vector (**E**) transfected A549 and H1975 cells pretreated with 20 μM nocodazole were stained for PCDHB13 (red) and α-Tubulin (white) at 0, 10, and 30 min after nocodazole washout. Higher magnifications of the areas in the white boxes marked by * or # are shown in the corresponding panel. Hoechst33342 was stained for nuclei (blue). Green arrows indicate MTOC with defective microtubule outgrowth. The spatial correlation of PCDHB13 with microtubules in the indicated region (pink lines mark two opposite ends, indicated by pink arrows) is shown by a line-series analysis. The red and white lines in the right panels represent the PCDHB13 and α-Tubulin fluorescence intensities, respectively. au: arbitrary units. Bar: 20 μm. (**F**) Quantification of MTOC formation in cells transfected by GFP or PCDHB13 after nocodazole washout. g Illustration of the phenomena shown in (**D**,**E**).

**Figure 5 cancers-11-00107-f005:**
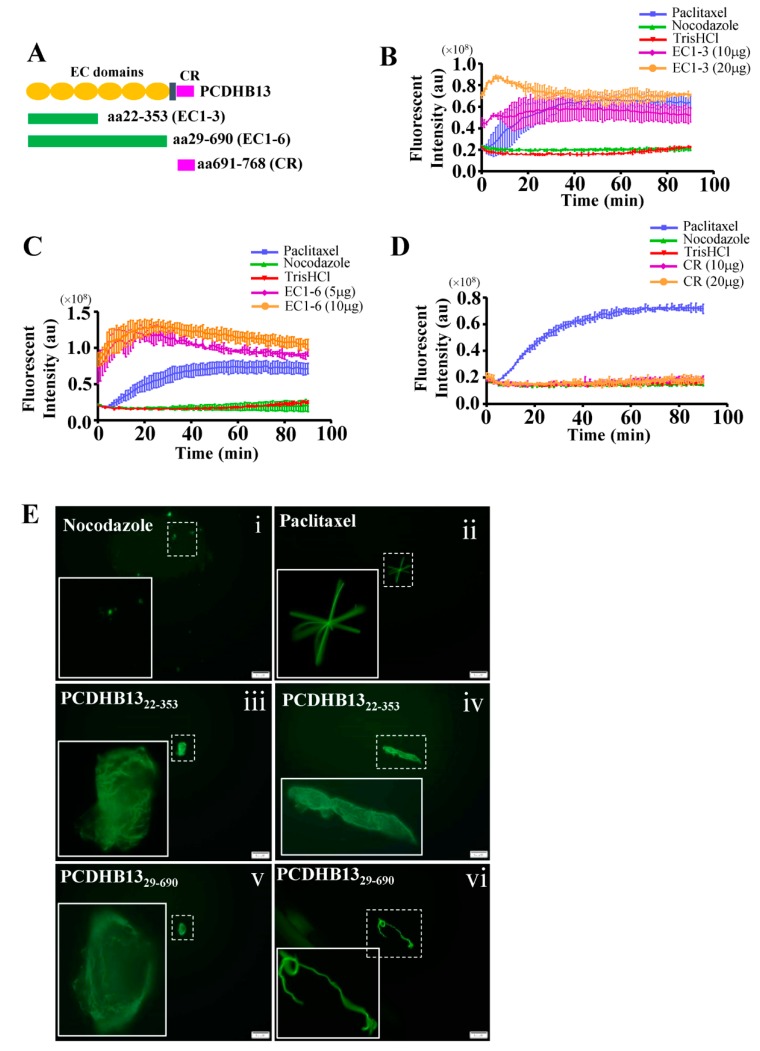
The extracellular domain of PCDHB13 regulates microtubule polymerization. (**A**) Schematic showing the functional domains of PCDHB13. Recombinant proteins with EC1–EC3, EC1–EC6, and CR of PCDHB13 were used for the in vitro tubulin polymerization assay. (**B**–**D**) Dynamic assessment of the effects of PCDHB13 domains on microtubule polymerization. Pure tubulins were assembled in the presence of indicated amounts of CR domain (**B**) or EC domains (**C**,**D**). Tris-HCl-10% DMSO, paclitaxel, and nocodazole served as vehicle, positive control, and negative control, respectively. (**E**) Fluorescent tubulins were used to probe the formation of microtubules in the presence of PCDHB13 EC domains. Boxed regions are magnified in the panels to the left. Bar: 20 μm. Three wells were analyzed for each independent replicate. Quantitative results are presented as the mean ± SD (*n* = 3).

**Figure 6 cancers-11-00107-f006:**
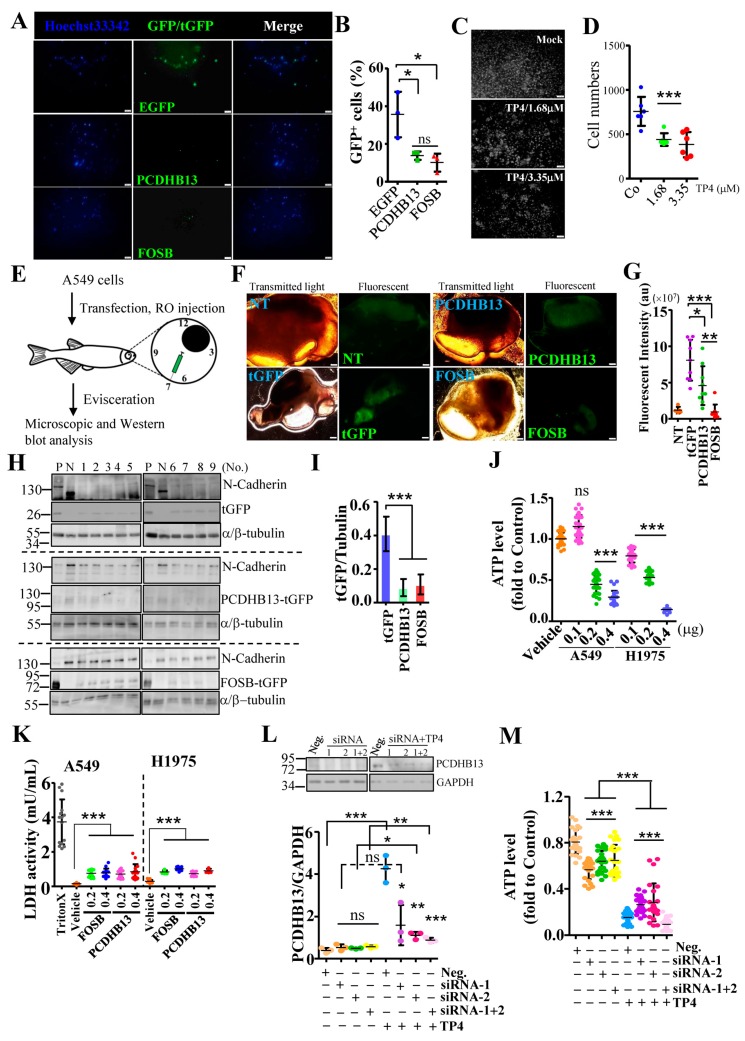
Elevated expression of PCDHB13 inhibits cell invasion and triggers cell death. (**A,C**) The underside of Matrigel-coated polycarbonate membranes used for cell invasion assays on EGFP-, PCDHB13-, and FOSB-transfected cells (**A**) or cells incubated with 1.68 and 3.35 μM TP4 (**C**). Cells were stained by Hoechst33342 shown in blue (**A**) or white (**C**). Bar: 200 μm. (**B**,**D**) Quantification of the cells that migrated across the membrane. Data were calculated by normalizing GFP/tGFP-positive to the total cell counts (*n* = 463 in EGFP transfected groups, *n* = 300 in PCDHB13-tGFP transfected groups, and *n* = 297 in FOSB-tGFP transfected group) in (**B**) or by counting Hoechst-dye stained cells in (**D**). (**E**) Schematic of the A549 cell xenotransplantation procedure. Cells were used for RO injection at 24 h post-transfection. Cell-transplanted zebrafish were cultured for 4 d. (**F**) Transmitted light and fluorescent images of A549 cells without or with transfection. Bar: 100 μm. (**G**) Quantification of fluorescent signal in dissected organs (*n* = 5 in the nontransfected/NT group and *n* = 9 in the transfected groups). au: arbitrary unit. (**H**) Tissue extracts from tGFP-, PCDHB13-, and FOSB-transfected A549-transplanted zebrafish were analyzed by Western blot using antibodies against N-Cadherin, α/β-Tubulin, and tGFP. P: Lysates from tGFP/PCDHB13/FOSB-transfected A549 cells. N: Lysate from nontransfected A549 cell. (**I**) Quantification of the EGFP/tGFP levels from Western blots (*n* = 9). (**J**) Cell viability was determined by the ATP assay for A549 and NCI-H1975 cells after transfection with PCDHB13. Ten replicate wells were analyzed for each dose (*n* = 3). (**K**) LDH release in A549 and NCI-H1975 cultures were determined 24 h after FOSB and PCDHB13 transfection. Triton-X was used as a positive control. Each independent replicate was measured at least in triplicate (*n* = 3). (**L**) Total lysates from A549 cells transfected with control (Neg.) or PCDHB13 siRNAs were analyzed by Western blot using antibodies against GAPDH and PCDHB13. Bottom, PCDHB13 levels were quantified and normalized to GAPDH. (**M**) Cell viability was determined by the ATP assay for A549 cells transfected with siRNAs targeting PCDHB13. Eight replicate wells were analyzed for each condition (*n* = 3). Quantitative results represent the mean ± SD (One-way ANOVA test in (**K**); two tailed *t*-test in (**M**): ns, not significant; * *p* < 0.05; ** *p* < 0.01; *** *p* < 0.001).

**Figure 7 cancers-11-00107-f007:**
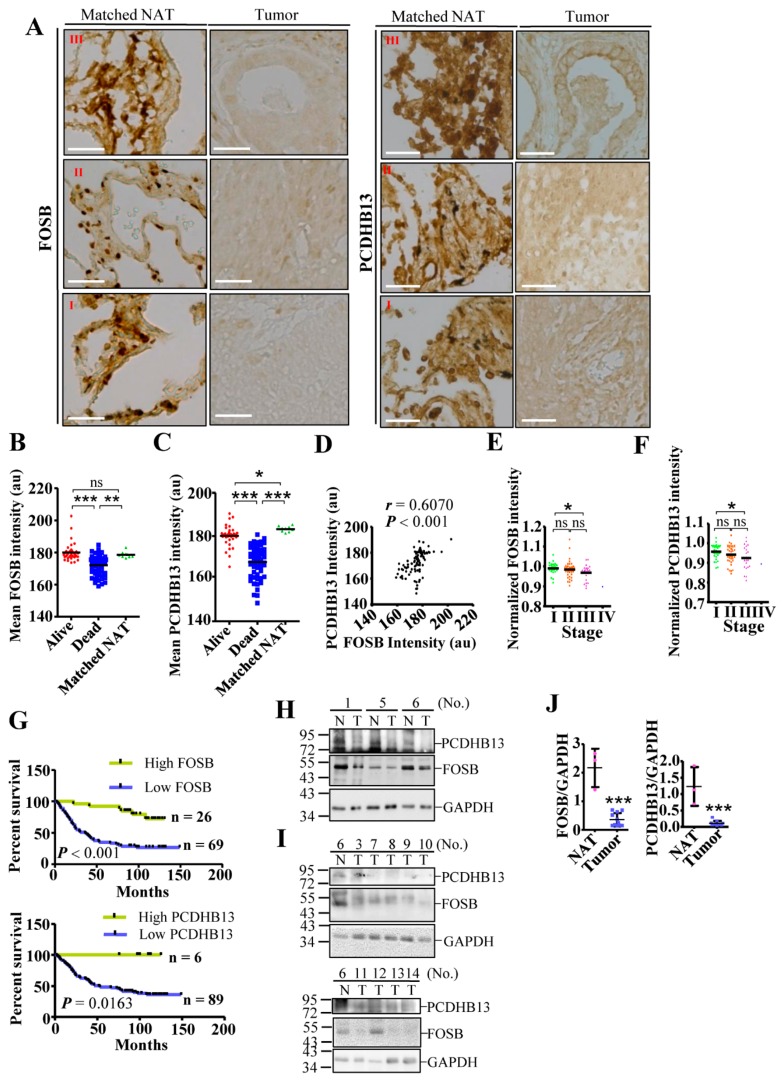
Downregulation of FOSB and PCDHB13 in lung cancer patients. (**A**) Matched NAT (*n* = 9) and lung cancer samples (*n* = 31, 42, 21, and 1 for grade I, II, III, and IV samples, respectively) were stained with DAB signal enhancement for FOSB and PCDHB13. Scale bar: 50 μm. (**B,C**) Quantification of DAB staining intensity for FOSB (**B**) and PCDHB13 (**C**) in alive, dead, and matched NAT samples. (**D**) Correlation between FOSB and PCDHB13 levels in the alive group (*p* < 0.001, Spearman’s *r* = 0.6070). (**E**,**F**) DAB staining intensity of FOSB (**E**) and PCDHB13 (**F**) were normalized to the respective values from matched NAT and were classified according to the disease stage. (**G**) Survival analysis of lung cancer patients according to relative FOSB expression (upper panel) and PCDHB13 expression (lower panel) level. Statistical comparisons of survival curves between groups were performed by Log-rank test. ** *p* < 0.001 for FOSB expression and **p* < 0.05 for PCDHB13 expression. (**H**,**I**) Tissue extracts from twelve NSCLC patients were analyzed by Western blot using antibodies against FOSB, PCDHB13, and GAPDH. P: Lysates from EGFP/PCDHB13/FOSB-transfected A549 cells. N: normal adjacent tissue; T: tumor. (**J**) Quantitation of the FOSB and PCDHB13 levels shown in (H) and (I) (*n* = 11). Quantitative results represent the mean ± SD (two-tailed *t*-test: ns, not significant; * *p* < 0.05; ** *p* < 0.01; *** *p* < 0.001).

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
