# Peer review of "FOSB–PCDHB13 Axis Disrupts the Microtubule Network in Non-Small Cell Lung Cancer"

_cancers, 2019, doi:10.3390/cancers11010107_

Reviewer 1 Report

The authors show a huge amount of data and many exiting discoveries in this study to support their conclusions including that TP4 treatment increases FOSB level and further regulates the downstream PCDHB13 which dysregulates tubulin strucutre in vitro and in vivo; high FOSB-PCDHB13 level inhibits NSCLC cells growth and migration. 

During reading the manuscript, I have some questions relative to the study:

1.     Whether TP4 increases FOSB at mRNA level or protein level?

2.     How mitochondrial functions affected by TP4 influence the transcriptional downstream events?

3.     If PCDHB13 dysregulates tubulin structure, whether TP4 treatment or overexpression of FOSB or PCDHB13 could arrest cells in mitosis? If yes, the cell cycle arrest effect could be another mechanism to reduce cell viability.

4.     About Figure 7, it is not clear about whether the IHC tissues were from NSCLC patients. It seems that the authors want to enlarge the range to the lung cancer in the manuscript. It raises the question of whether the low FOSB and PCDHB3 expression correlates with some specific lung cancer types (for example, NSCLC) according to some regulatory mechanisms and the low FOSB and PCDHB3 group of patients correlate with a poor therapeutic survival data comparing to the high FOSB and PCDHB3 group. However, the data in Figure 7 is not solid enough to this point. Please tone down in the text, for example, Line 342, “might be associated with…”

5.     The quality of some western blots in this study is worrisome, for example, Fig 1A BEAS-2B FOSB blot and some PCDHB13 blots. A better antibody for blots or confirming the phenomenon on mRNA level by RT-qPCR could help to make solid conclusions.

There are also some minor problems:

1.     Line 75-76, it is better to include some detailed cell lines information for readers, for example, “in all four NSCLC cell lines tested but not in the control, a normal human bronchial epithelial cell line, BEAS-2B.” Or include this information in the figure legend.

2.     For Figure 1B, it is not a clear graph to overlap the two sets of data of FOSB and FOSΔB. It is better to separate them into different dot groups or into different graphs.

3.     Line 143, it should be “Supplementary Figure S4Di, ii”.

4.     Line 161, the summary of Figure 2 is not precise. Only based on transcriptome GO analysis, there was not any more data supporting effects on “membrane intergrity”. 

5.   In Figure 2F, please make it clear in the figure legends what is “C” treatment (Control?). Or is it a mistaken “M” (Mock)? 

The authors should carefully check the rest of the manuscript to correct similar problems.

Author Response

Response to Reviewer 1 Comments

During reading the manuscript, I have some questions relative to the study:

1. Whether TP4 increases FOSB at mRNA level or protein level?

Ans: We thank the reviewer for this comment. We discovered that FOSB is induced by TP4 through a transcriptomic analysis of breast cancer cells, suggesting FOSB expression is enhanced at the mRNA level (Ref. 14). In response to the reviewers’ question, we have performed RT-qPCR to determine FOSB expression at mRNA level. The result shows that FOSB mRNA level gradually increases post-TP4 treatment (see the figure below) and is consistent the the gradual increase of FOSB protein level shown in Figure 1K. This result suggests that the increase of FOSB protein level by TP4 in lung cancer cells is through the activation of FOSB gene.                                         

2. How mitochondrial functions affected by TP4 influence the transcriptional downstream events?

Ans: We thank the reviewer for raising this critical point. We observed intraceullular penetration of TP4, which caused mitochondrial pathology and led to the dysregulation of Ca2+-homeostasis. We show that the TP4-mediated induction of FOSB (Figure 1L) and cell death (Figure 1M) both required Ca2+, indicating Ca2+ plays a central role in the regulation of downstream transcriptional events. AP-1 family transcription factors are immediate early genes (IEGs) expressed in response to cellular stresses and are induced by Ca2+ agonist in various cell types (Roche E and Prentki M., Cell Calcium. 1994,16:331-8). In addition, elevated extracellular Ca2+ in cultured cells has been shown to enhance gene expression (Ng DC. et al., J Biol Chem. 2000, 275:24080-8). A Ca2+ response element (CRE) containing an AP-1 site was identified in the upstream region of the involucrin gene promoter. Mutation of the sequence abolished the promoter activation by Ca2+. Moreover, Ca2+ treatment enhanced the levels of JUN family (c-JUN, JUNB, JUND) and FOS family (FOSB and FRA-1) proteins, suggesting that Ca2+-regulated involucrin gene expression is mediated at least in part by AP-1 transcription factors. However, the question of whether any calcium-response element exists in the FOSB promoter region has not yet been evaluated.

3. If PCDHB13 dysregulates tubulin structure, whether TP4 treatment or overexpression of FOSB or PCDHB13 could arrest cells in mitosis? If yes, the cell cycle arrest effect could be another mechanism to reduce cell viability.

Ans: We thank the review for this comment and very much agree with this point of view. Indeed, we have previously characterized TP4 as a novel microtubule destabilizing agent (Ting CH, et al., Mar Drugs. 2018, 16(12), pii: E462, Ref.15). Like other microtubule binding agents (MBAs, e.g., vinca alkaloids), TP4 treatment caused obvious chromatin condensation (Figure 1C form Ref.15) in A549 cells, suggesting mitotic suppression may be another cellular effect caused by TP4. However, like other MBAs, TP4 may trigger cancer cell death through a very different mechanism. TP4-tubulin binding is expected to disrupt lateral interactions between tubulin protofilaments and to compete with the interaction between tubulin and microtubule-binding proteins (Ref.15). However, taxnes (e.g., docetaxel and paclitaxel) bind to the β-subunit of microtubules and ‘fix’ the tubulin to prevent disassembly (Nogales, E., et al., Nature 1995, 375, 424-427; Rao, S., et al., J. Biol. Chem. 1994, 269, 3132-3134). Most MBAs cause apoptoic cell death through mitotic supression, while TP4 has been shown to dominantly cause necrotic cell death in different cancer types (Ref. 14,15,17). We also found that TP4 induced strong FOSB expression in breast cancer cells, but this effect was not observed in cells treated with other MBAs, such as docetaxel and paclitaxel (Ref. 14). These findings raise the possibility that the specific manner by which microtbules are targeted may determine which cellular stress responses are induced. Interestingly, both microtubule and mitochondria are intracellular targets for the TP4. Whetehr and how these TP4-induced cellular stress responses may work synergistically to initiate death signaling in different types of cancer cells remains to be further studied.

4. About Figure 7, it is not clear about whether the IHC tissues were from NSCLC patients. It seems that the authors want to enlarge the range to the lung cancer in the manuscript. It raises the question of whether the low FOSB and PCDHB3 expression correlates with some specific lung cancer types (for example, NSCLC) according to some regulatory mechanisms and the low FOSB and PCDHB3 group of patients correlate with a poor therapeutic survival data comparing to the high FOSB and PCDHB3 group. However, the data in Figure 7 is not solid enough to this point. Please tone down in the text, for example, Line 342, “might be associated with…”

Ans: We thank the reviewer for this comment. We have corrected the text in the Results section (line 361).

5. The quality of some western blots in this study is worrisome, for example, Fig 1A BEAS-2B FOSB blot and some PCDHB13 blots. A better antibody for blots or confirming the phenomenon on mRNA level by RT-qPCR could help to make solid conclusions.

Ans: We thank the reviewer for this comment. The BEAS-2B blot in Figure 1A (and the uncropped image shown in Figure S1) has been replaced by a new blot using the FOSB antibody. In addition, the blot probed with PCDHB13 antibody in Figure 3A, Figure 4C, and Figure 6L were replaced by the same data with longer exposure time (Figure 3A as recommended by the reviewer 2) or by repeating the experiments using the same samples (Figure 4C and 6L).

There are also some minor problems:

1. Line 75-76, it is better to include some detailed cell lines information for readers, for example, “in all four NSCLC cell lines tested but not in the control, a normal human bronchial epithelial cell line, BEAS-2B.” Or include this information in the figure legend.

Ans: We thank the reviewer for this suggestion. We have included ²a normal human bronchial epithelial cell line, BEAS-2B² in the figure legend (line 85). In addition, four ²NSCLC cell-lines² was corrected to four ²lung epithelial cancer cells² (line 84-85).

2. For Figure 1B, it is not a clear graph to overlap the two sets of data of FOSB and FOSΔB. It is better to separate them into different dot groups or into different graphs.

Ans: We thank the reviewer for this comment. We have separate the datasets for FOSB and FOSDB overexpression in Figure 1B.

3. Line 146, it should be “Supplementary Figure S4Di, ii”.

Ans: We thank the reviewer for pointing this error. It has been corrected (line 159).

4. Line 161, the summary of Figure 2 is not precise. Only based on transcriptome GO analysis, there was not any more data supporting effects on “membrane integrity”.

Ans: We thanks the reviewer for this comment. The description of ²membrane integrity² was deleted (line 176).

5. In Figure 2F, please make it clear in the figure legends what is “C” treatment (Control?). Or is it a mistaken “M” (Mock)? The authors should carefully check the rest of the manuscript to correct similar problems.

Ans: We thank the reviewer for this comment. We have corrected the ²M² throughout the manuscript to ²Co² (Control treatment) (lines 119, 131, 138, 148, 182, 625, 629, 630, and 634), Figures, and Supplementary Figures.

Reviewer 2 Report

The authors of the manuscript characterize an interesting antimicrobial peptide that induces FOSB as well as its downstream target PCDHB13. Through in vitro functional assays they identify that these signaling cascades may regulate some cellular functions in NSCLC cell lines including apoptosis and invasion. Furthermore, they show in vivo that knockdown of FOSB and PCDHB13 inhibits NSCLC cell line proliferation in a zebrafish model. While the methods and data appear to be scientifically sound, I had some reservations regarding the manuscript.

There are several grammatical errors as well as revisions in verbiage that need to be made in order to read correctly. (i.e. line 107 "(C)" or line 149 "induced," etc.)

It was not readily explained as to why TP4 had an effect on cancer cells however, had no effect on the BEAS2B "normal" epithelial cell line. Is this due to mutations present in cancers that cause a response? 

Figure 1G - line 86 described changes in "morphology." Perhaps there is a better way to phrase these results. I generally think of changes in morphology as being depicted by microscopy not western blot analysis.

Figure 1A - Is there a clearer image for the BEAS2B cell line showing FOSB - the image is hard to discern clear protein bands

Figure 1E - The x-axis showing treatment groups does not line up with the figure

Figure 1 - It is not explained as to why the concentrations (0.84, 1.68, 6.71 uM) are used

It is not described why the proteins Prohibitin, Giantin, and Calreticulin were labeled in Figure 1. Perhaps a sentence describing the function of each protein in the context of the findings would be appropriate.

The format for defining the number of replicates and independent experiments performed changes between Figures 1 (at end of legend) and Figure 5 (after each panel description). It is more appropriate to keep this consistent throughout.

Figure 2C - usually a decrease in E-Cadherin is present with an increase in N-Cadherin in the context of EMT. Can you describe why you see a descrease in protein levels of both E and N-Cadherin?

Several images taken for western blot analysis show proteins at the same size within the same panel. Was the same membrane stripped and reprobed with different antibodies? Were the images resulting from membranes run in duplicate? These methods are not described in the materials and methods section.

There was no data representing knockdown efficiency when using siRNA-mediated mRNA k.d. It may be helpful to identify the transfection efficiency in your cell lines. Additionally, qRT-PCR analysis would provide some insight into whether or not the genes targeted by the siRNAs are significantly knocked down.

Does TP4 enter the cancer cells? Does it interact with any extracellular receptors? Is this a completely extrinsic response that might be mediated through some receptor-mediated mechanism?

Figure 2G - Were the siRNA tested individually for an effect or pooled from the start of the experiment? Did both show an effect when transfected into cells individually?

Figure 2 - line 174 - Should read (D,F,H) -- I believe.

Figure 3A - It is hard to see any bands apart from Neg-si + TP4. Is there an image in which these bands were exposed for a longer time?

Figure 3D - The labels for treatment groups in offset and makes reading the results difficult

Figure 3H - The exposure setting for mCherry appear inconsistent between H (top) and (bottom). 

Line 213 - These results were not generated from an in vivo experiment -- cell lysates from in vitro

Figure 4A - there are no pink arrows as described in the legend, only red are shown on this panel

Figure 4G is not described or mentioned at all in the legend

Figure 4C is difficult to read and tubulin appears inconsistently loaded between lanes

Line 279 - RO - retro orbital needs to be spelled out before using abbreviations

line 285 "tumorigenesis" could be replaced with cell proliferation -- the cells are already tumorigenic

Figure 6H - it is not explained as to why N-Cadherin was analyzed by western blot. It may also be a good idea to leave tubulin at the bottom and eGFP in the middle for all transfection groups.

Figure 6L - Why does the siRNA used to knockdown PCDHB13 appear to be inducing it in the western blot analysis? This is not seen when TP4 is added in combination with siRNA.

Additionally, in Panel M the siRNAs appear to antagonize each other when looking at cell viability, but again this effect disappears with the addition of TP4. How can this be better explained in the discussion? Is this because the loading of protein does not appear to be consistent (looking at GAPDH bands between TP4 - and TP4 + groups)?

Line 312 - cells were not transfected with PCDHB13 -- change to "siRNAs targeting PCDHB13"

One question I have is the fact that several studies have shown FOSB and certainly AP-1 to be overexpressed in cancers. How do these findings fit with current literature which identify FOSB as a proto-oncogene?

Figure 7G - Were these patients matched on age, gender, smoking status, etc? There may be potential confounding present which could bias the results.

It would be nice to see experiments in which calcium dysregulation (shown in Figure 1) was tied into the effects seen in later Figures including cytoskeletal remodeling. Perhaps a quick rescue experiment could determine whether or not Ca is necessary for certain TP4-induced effects through BFOS to occur?

Supplemental Figure 1J - uses C and T instead of M and T -- Be consistent with labeling of treatment groups in all Figures

Author Response

Response to Reviewer 2 Comments

There are several grammatical errors as well as revisions in verbiage that need to be made in order to read correctly. (i.e. line 107 "(C)" or line 149 "induced," etc.)

Ans: We thank the reviewer for this comment. We have corrected the grammatical errors throughout the manuscript and figures. The ²C² and ²M” were corrected to be ²Co² for the control treatment (line 119, 131, 138, 148, 182, 625, 629, 630, and 634). ²TP4-induced² was corrected as ²TP4-caused² in the revised manuscript.

It was not readily explained as to why TP4 had an effect on cancer cells however, had no effect on the BEAS2B "normal" epithelial cell line. Is this due to mutations present in cancers that cause a response?

Ans: We thank the reviewer for this comment. The selective cancer-killing ability of cationic AMPs is derived from a structural amphipathic property, which enables electrostatic interactions with anionic molecules on the plasma membrane of cancer cells, but this anionic nature is absent in normal cells (Ref.12). We have included some references and the following description into the Introduction section: ² AMPs have been reported to selectively target cancer cells with lower toxicity to normal cells and thus may serve as potent anti-cancer drugs [12-15]. The selective toxicity to cancer cells is probably due to the negative charge often found ond the plasma membranes of cancer cells [12, 14, 16]. Strong cytotoxicity of TP4 was previously observed in NSCLC cells but not in normal cells [17]. Extensive glycosylation and low cholesterol levels are found on lung cancer cell membranes are not found in normal lung cells and may enhance the negative charge of the plasma membrane on the cancer cells [18, 19]² (line 44-50).

Figure 1G - line 86 described changes in "morphology." Perhaps there is a better way to phrase these results. I generally think of changes in morphology as being depicted by microscopy not western blot analysis.

Ans: We thank the reviewer for this comment. We have rephrased the description and corrected some labeling errors (line 96-101).

Figure 1A - Is there a clearer image for the BEAS2B cell line showing FOSB - the image is hard to discern clear protein bands

Ans: We thank the reviewer for this comment. The BEAS-2B blot in Figure 1A (and the uncropped image shown in Figure S1) have been replaced by a new blot using the FOSB antibody.

Figure 1E - The x-axis showing treatment groups does not line up with the figure

Ans: We thank the reviewer for pointing out this error. We have corrected the axis in Figure 1E.

Figure 1 - It is not explained as to why the concentrations (0.84, 1.68, 6.71 mM) are used

Ans: We thank the reviewer for this comment. The doses of TP4 were selected based on our earlier work showing that concentrations over 3.35 mM caused cytotoxicity to the NSCLC cells. We have added details related to the doses of TP4 in Figure 1 (line 80-81, 100-101, and 103-104).

It is not described why the proteins Prohibitin, Giantin, and Calreticulin were labeled in Figure 1. Perhaps a sentence describing the function of each protein in the context of the findings would be appropriate.

Ans: We thank the reviewer for this comment. We have included descriptions of the organelle markers we used in the manuscript (line 92-95).

The format for defining the number of replicates and independent experiments performed changes between Figures 1 (at end of legend) and Figure 5 (after each panel description). It is more appropriate to keep this consistent throughout.

Ans: We thank the reviewer for this comment. We have made correction in the Figure 5 to keep the format consistent (line 284-285).

Figure 2C - usually a decrease in E-Cadherin is present with an increase in N-Cadherin in the context of EMT. Can you describe why you see a decrease in protein levels of both E and N-Cadherin?

Ans: We thank the reviewer for this comment. In some cancer cells, EMT may be accompanied by the decrease of E-Cadherin and/or the increase of N-Cadherin. Cadherin molecules are required for the maintenance of the highly dynamic adherens junctions. These cadherins turn over rapidly. The microtubule cytoskeleton is critical for the endocytosis of cadherin molecules during normal and EMT conditions (Mary S., et al., Mol Biol Cell. 2002, 13:285-301; Benjamin A. N., et al., Subcell Biochem. 2012; 60: 197–222). In addition, cadherin expression is linked with mitochondrial energy production in cancer cells (Park S. Y., Cancer Sci. 2017, 108:1769-1777). Higher protein turnover rate must have considerable biological importance for so much energy to be spent. However, it remains unclear whether mitochondrial defects or cytoskeletal damage can affect cadherin levels. Overall, we propose that TP4 disrupts the cytoskeleton and causes cellular stress in lung cancer cells, which leads to the disruption of Cadherin regulation at protein and/or gene level.

Several images taken for western blot analysis show proteins at the same size within the same panel. Was the same membrane stripped and reprobed with different antibodies? Were the images resulting from membranes run in duplicate? These methods are not described in the materials and methods section.

Ans: We thank the reviewer for this comment. We have added the description: ² Protein samples were loaded in duplicate gels and transferred to separate membranes for probing with different antibodies. ² in the Methods section (line 519-521).

There was no data representing knockdown efficiency when using siRNA-mediated mRNA k.d. It may be helpful to identify the transfection efficiency in your cell lines. Additionally, qRT-PCR analysis would provide some insight into whether or not the genes targeted by the siRNAs are significantly knocked down.

Ans: We thank the reviewer for this comment and entirely agree with the point. Because of the deadline for revision and the time spent waiting for the back-ordered siRNAs to arrive, we are not able to perform these experiments. However, we have observed that the FOSB or PCDHB13 siRNAs (siRNA-1, -2 or used in combination) we used in this study can reach to over 75% knockdown efficiency when compared with the negative control siRNA KD group upon TP4 treatment (Figures 1C and 6L). Therefore, we believe that these siRNAs worked efficiently in these experiments.

Does TP4 enter the cancer cells? Does it interact with any extracellular receptors? Is this a completely extrinsic response that might be mediated through some receptor-mediated mechanism?

Ans: We thank the reviewer for this question. We observed that TP4 is a cell-penetrating peptide (Ref. 14, 15). We did not identify an extracellular receptor which may interact with TP4 using a pull-down assay with TP4 antibodies and LC-MS/MS (Ref. 15). However, several intracellular targets have been characterized, including a-tubulin and an unpublished mitochondrial membrane protein. These findings may provide an answer for the observations that TP4 disrupted microtubule cytoskeleton and mitochondrial pathology.

Figure 2G - Were the siRNA tested individually for an effect or pooled from the start of the experiment? Did both show an effect when transfected into cells individually?

Ans: We thank the reviewer for this question. The cell lysates used in Figure 2G were the same as those used in Figure 1C, for which FOSB knockdown was confirmed. Because the knockdown efficiency of FOSB si-1+2 seemed to be better than individual siRNAs in A549 cells, we subsequently only used the lysates from the combined siRNA-treated group for later Western blot experiments.

Figure 2 - line 174 - Should read (D, F, H) -- I believe.

Ans: We thank the reviewer for this comment. We have deleted redundant sentences and corrected as ²Quantitative measurements of protein levels were normalized to GAPDH in (D, F, H)² (line 185-186).

Figure 3A - It is hard to see any bands apart from Neg-si + TP4. Is there an image in which these bands were exposed for a longer time?

Ans: We thank the reviewer for this point. We have replaced the image with another one with longer exposure time (Figure 3A).

Figure 3D - The labels for treatment groups in offset and makes reading the results difficult.

Ans: We thank the reviewer for pointing this error, we have corrected the labeling in Figure 3D.

Figure 3H - The exposure setting for mCherry appear inconsistent between H (top) and (bottom).

Ans: We thank the reviewer for this comment. We have made the exposure settings consistent in Figure 3H.

Line 213 - These results were not generated from an in vivo experiment -- cell lysates from in vitro

Ans: We thank the reviewer for this comment. We have deleted ²in vivo² from the sentence (line 227).

Figure 4A - there are no pink arrows as described in the legend, only red are shown on this panel

Ans: We thank the reviewer for this comment. The cellular aggregates were labeled by white circles in each of the sub panels shown in figure 4A (line 259).

Figure 4G is not described or mentioned at all in the legend

Ans: We thank the reviewer for this comment. We have added a statement ²A proposed model for the PCDHB13-driven microtubule disruption is shown in Figure 4G² at line239-240.

Figure 4C is difficult to read and tubulin appears inconsistently loaded between lanes

Ans: We thank the reviewer for this comment. The experiment was repeated using the same samples for Western blot. The protein concentration of the samples from the IP sample (Lane 4) may be different from the input cell lysates (lane 1 or lane 2), and thus the tubulin signal appears inconsistent.

Line 279 - RO - retro orbital needs to be spelled out before using abbreviations

Ans: We thank the reviewer for pointing this error. We have defined retro-orbital (RO) at line295.

line 285 "tumorigenesis" could be replaced with cell proliferation -- the cells are already tumorigenic

Ans: We thank the reviewer for this suggestion. We have deleted ²tumorigenesis² and added ²cell proliferation² (line301-302).

Figure 6H - it is not explained as to why N-Cadherin was analyzed by western blot. It may also be a good idea to leave tubulin at the bottom and eGFP in the middle for all transfection groups.

Ans: We thank the reviewer for this suggestion. (1) The N-Cadherin antibody was used to detect xenotransplanted cells. We have stated this point at line299. (2) Figure 6H has been modified in accordance with the suggestion.

Figure 6L - Why does the siRNA used to knockdown PCDHB13 appear to be inducing it in the western blot analysis? This is not seen when TP4 is added in combination with siRNA. Additionally, in Panel M the siRNAs appear to antagonize each other when looking at cell viability, but again this effect disappears with the addition of TP4. How can this be better explained in the discussion? Is this because the loading of protein does not appear to be consistent (looking at GAPDH bands between TP4 - and TP4 + groups)?

Ans: We thank the reviewer for this comment. Because cell transfection and TP4 treatment show some toxicity to cells, we collected the cell lysates by pooling multiple wells of cultured cells (triplicate wells from a 12-well plate for each treatment and repeated in three independent assays). The protein concentration in each group was quantitated before Western blotting. The results are shown in the Figure 6L and in the Supplementary Figure S9. In panel M, the siRNA (si-1 and si-2)-treated groups show relatively bad knockdown ability compared to the pooled treated (si-1+2) group (there is no statistically significant difference by unpaired t-test, Figure 6L), and thus relatively better viability is observed in the si-1 and si-2 treatment groups.

Line 312 - cells were not transfected with PCDHB13 -- change to "siRNAs targeting PCDHB13"

Ans: We thank the reviewer for pointing this error. We have made the correction at line 336.

One question I have is the fact that several studies have shown FOSB and certainly AP-1 to be overexpressed in cancers. How do these findings fit with current literature which identify FOSB as a proto-oncogene?

Ans: We thank the reviewer for this critical question. Indeed, FOSB was found as a proto-oncogene that can transform osteoblast cell as early as 20 years ago. However, evidence has also been provided that AP-1 family members (i.e. FOS and JUN) work as a double edged sword in tumorigenesis (Ref.16). AP-1 transcription factors can suppress tumor formation, depending on the cancer cell types, differentiation states, stages, and the genetic background. In this work and our previous studies, we observed that antimicrobial peptide (AMP) treatment (e.g., TP4) elevates expression of FOSB in cancers. In the cancers we tested, FOSB is absent in the cancerous tissues (i.e., breast and lung cancer) but is still expressed at the normal tissue. This finding suggested that FOSB has a role in normal cells but the expression is silenced during tumor formation. Indeed, FOSB expression was previously detected in well-differentiated breast tissues but was found to be absent during breast cancer progression (Ref.18 and 19). In addition, FRA-1, another FOS family member, is upregulated during breast cancer formation (Ref.14). This phenomenon suggested a molecular switch between the JUN/FOSB to JUN/FRA-1 in the breast cancers. However, we did not observe that FRA-1 level was changed in NSCLC cells (Figure S1J). Based on the current knowledge that AP-1 activity is highly regulated in some cellular contexts, it would be of great interest to characterize the expression of each AP-1 member between disease stages and cancer cell status (i.e., poorly or highly differentiation).

Figure 7G - Were these patients matched on age, gender, smoking status, etc? There may be potential confounding present which could bias the results.

Ans: We thank the reviewer for this comment. Indeed, these factors mentioned by the reviewer significantly affect the disease status in lung cancer. We have used the on-line Kaplan-Meier plotter (http://kmplot.com/analysis/) to analyze the transcriptomic database based on the gender and smoking status. Only the results with significant differences (log-rank P < 0.05) are shown as the Supplementary Figure S7A-D in our revised manuscript. Higher overall survival rate is observed in male non-smokers with higher FOSB level, male smokers and male/female non-smokers with higher PCDHB13 level. Information about the twelve NSCLC patients are also provided in the Supplementary Figure S7E. Further epidemiologic studies may be performed to continue to unravel the complexities of these biological factors in NSCLC.

It would be nice to see experiments in which calcium dysregulation (shown in Figure 1) was tied into the effects seen in later Figures including cytoskeletal remodeling. Perhaps a quick rescue experiment could determine whether or not Ca is necessary for certain TP4-induced effects through FOSB to occur?

Ans: We thank for the reviewer for this comment. We have performed a rescue experiment by treating cells with a Ca2+ chelator prior to TP4 and followed by analyzing FOSB as well as microtubule remodeling by ICC. However, this experimental design is not working because treatment of cells with the BAPTA/AM strongly disrupts microtubule dynamics (see the figure below). Because we need to revise our manuscript in a timely manner, we are afraid that we are not able to conduct further experiments to address this issue at the current stage.

Supplemental Figure 1J - uses C and T instead of M and T -- Be consistent with labeling of treatment groups in all Figures

Ans: We thank the reviewer for this comment. We have corrected the errors throughout the manuscript and figures. The ²C² and ²M” were corrected as ²Co² for the control treatment (line 119, 131, 138, 148, 182, 625, 629, 630, and 634).
